# UHPLC/MS-Based Serum Metabolomics Reveals the Mechanism of Radiation-Induced Thrombocytopenia in Mice

**DOI:** 10.3390/ijms23147978

**Published:** 2022-07-20

**Authors:** Ling Xiong, Long Wang, Ting Zhang, Xinyuan Ye, Feihong Huang, Qianqian Huang, Xinwu Huang, Jianming Wu, Jing Zeng

**Affiliations:** 1School of Pharmacy, Southwest Medical University, Luzhou 646000, China; 20200599120109@stu.swmu.edu.cn (L.X.); wanglong1226@swmu.edu.cn (L.W.); 20200599120042@stu.swmu.edu.cn (T.Z.); 20200599120078@stu.swmu.edu.cn (X.Y.); huangfeihong@swmu.edu.cn (F.H.); huangqianqian@swmu.edu.cn (Q.H.); huangxinwu2021@swmu.edu.cn (X.H.); 2Education Ministry Key Laboratory of Medical Electrophysiology, Key Medical Laboratory of New Drug Discovery and Druggability Evaluation, Key Laboratory of Activity Screening and Druggability Evaluation for Chinese Materia Medica, Southwest Medical University, Luzhou 646000, China

**Keywords:** radiation-induced thrombocytopenia, metabolomics, UHPLC-QTOF MS, L-tryptophan, LysoPC (17:0), D-sphinganine

## Abstract

Radiation-induced thrombocytopenia is a common and life-threatening side effect of ionizing radiation (IR) therapy. However, the underlying pathological mechanisms remain unclear. In the present study, irradiation was demonstrated to significantly reduce platelet levels, inhibit megakaryocyte differentiation, and promote the apoptosis of bone marrow (BM) cells. A metabolomics approach and a UHPLC-QTOF MS system were subsequently employed for the comprehensive analysis of serum metabolic profiles of normal and irradiated mice. A total of 66 metabolites were significantly altered, of which 56 were up-regulated and 10 were down-regulated in irradiated mice compared to normal mice on day 11 after irradiation. Pathway analysis revealed that disorders in glycerophospholipid metabolism, nicotinate and nicotinamide metabolism, sphingolipid metabolism, inositol phosphate metabolism, and tryptophan metabolism were involved in radiation-induced thrombocytopenia. In addition, three important differential metabolites, namely L-tryptophan, LysoPC (17:0), and D-sphinganine, which were up-regulated in irradiated mice, significantly induced the apoptosis of K562 cells. L-tryptophan inhibited megakaryocyte differentiation of K562 cells. Finally, serum metabolomics was performed on day 30 (i.e., when the platelet levels in irradiated mice recovered to normal levels). The contents of L-tryptophan, LysoPC (17:0), and D-sphinganine in normal and irradiated mice did not significantly differ on day 30 after irradiation. In conclusion, radiation can cause metabolic disorders, which are highly correlated with the apoptosis of hematopoietic cells and inhibition of megakaryocyte differentiation, ultimately resulting in thrombocytopenia. Further, the metabolites, L-tryptophan, LysoPC (17:0), and D-sphinganine can serve as biomarkers for radiation-induced thrombocytopenia.

## 1. Introduction

With the widespread application of nuclear technology in clinical and daily life, and the frequent occurrence of nuclear accidents, people are more frequently experiencing injuries caused by irradiation. Ionizing radiation, the main type of radiation, includes X-ray, γ-ray, β-ray, α-ray, and rapid neutron radiation [1,2]. Among these, X-ray radiation is widely used to treat various malignant tumors and cancers [3]. However, ionizing radiation therapy is associated with several side effects, including gastrointestinal reactions, bone marrow suppression, and thrombocytopenia. Among these, thrombocytopenia is an important cause of morbidity and mortality in acute radiation syndrome [4,5]. Platelets not only play an important role in the regulation of hemostasis and thrombosis, but also in innate immunity and the regulation of tumor growth and extravasation [6]. Platelet levels have been reported to be associated with the prognosis of diseases [7,8,9]. In fact, the mean platelet volume (MPV) and MPV/PLT ratio (MPR) can be used to predict 28-day mortality in patients with COVID-19 pneumonia [10]. According to some studies, thrombocytopenia can lead to bleeding and clotting disorders and can be life threatening [11].

Ionizing radiation causes both direct and indirect damage. On one hand, radiation can directly cause DNA damage, which leads to genetic mutations and cancer. On the other hand, radiation interacts with organs, resulting in the production of various oxygen free radicals, especially free radicals such as H• and •OH, which lead to the production of toxic reactive oxygen species (ROS), such as superoxide radical (•O_2_^−^) and hydrogen peroxide (H_2_O_2_) [3,12]. ROS play an important role in the injury of proteins, lipids, and DNA, and can lead to different diseases including cancer [13]. The hematopoietic system is one of the systems most sensitive to radiation. Both short-term high-dose and long-term low-dose radiation exposure can cause radiation damage in the hematopoietic system [14]. Myelosuppression is the main cause of radiation injury of hematopoietic system. Bone marrow (BM) injury has been proven to be one of the important processes involved in acute radiation syndrome [15,16]. Radiation-induced injury to the hematopoietic system is largely due to the extreme sensitivity of BM cells to genotoxic stress [17]. Damage to and the death of megakaryocytes induced by ionizing radiation can cause a decrease in platelet production [18], as platelets are produced by megakaryocytes in the BM. Previously, radiation exposure was demonstrated to induce the accumulation of DNA damage, which leads to the apoptosis of BM cells [19]. Further, radiation causes hematopoietic progenitor/stem cells to lose their stemness [20]. However, whether radiation-induced thrombocytopenia is associated with abnormalities in body metabolism remains unclear.

Metabolomics is a branch of the omics techniques that explores the regulatory mechanism of metabolic function by detecting the metabolic map of the entire body [21]. Metabolomics can help identify potential biomarkers and dysfunctional metabolic pathways by comparing metabolic profiles in normal and diseased states, ultimately leading to the interpretation of possible mechanisms [22]. The most commonly-used metabolomics techniques include liquid chromatography-mass spectrometry (LC-MS) [23,24,25], gas chromatography-mass spectrometry (GC-MS) [26,27], and nuclear magnetic resonance (NMR) [28,29]. Ultra-high-performance liquid chromatography tandem mass spectrometry (UHPLC-MS), one of the most widely used metabolomics technologies, can detect large quantities of metabolic characteristics from biological samples with high throughput and sensitivity [30]. Amrita et al. used metabolomics to identify intestinal pathological biomarkers by detecting changes in endogenous intestinal substances in mice exposed to long-term radiation [31]. Zhao et al. screened radiation-sensitive biomarkers in rat plasma using metabolomics [32]. Recently, studies on the radiation protective effect of traditional Chinese medicine (TCM) based on metabolomics have gradually increased [2,33]. According to previous studies, the fate of hematopoietic stem cells (HSCs) is controlled by metabolism, and many hematological diseases are related to abnormal metabolic patterns [34]. Therefore, the association between abnormal metabolic patterns and radiation-induced thrombocytopenia requires further investigation, and the mechanisms of action of the subsequently altered small metabolites need to be further studied.

In the present study, untargeted metabolomics and multivariate statistical analysis were combined to elucidate the mechanism of radiation-induced thrombocytopenia in mice. As a result, the metabolic pattern of irradiated mice was found to significantly change compared to that of normal mice on day 11 after irradiation. Further, 66 differential metabolites corresponding to glycerophospholipid metabolism, nicotinate and nicotinamide metabolism, sphingolipid metabolism, inositol phosphate metabolism, and tryptophan metabolism pathways were identified. The effects of three important differential metabolites, L-tryptophan, LysoPC (17:0), and D-sphinganine on megakaryocyte differentiation and apoptosis were also verified. Based on the results, L-tryptophan, LysoPC (17:0), and D-sphinganine-induced apoptosis, and L-tryptophan inhibited the megakaryocyte differentiation of K562 cells. Once the platelet level in irradiated mice returned to normal levels, the contents of L-tryptophan, LysoPC (17:0), and D-sphinganine recovered to normal levels. Overall, our findings highlight the underlying mechanism of radiation-induced thrombocytopenia and reveal potential methods for prevention on metabolic signaling pathways in the treatment of thrombocytopenia.

## 2. Results

### 2.1. Influence of Radiation on Hematopoiesis in Mice

Thirty-two KM mice were randomly divided into two groups (eight males and eight females in each group): a normal group and a radiation group. To investigate the effect of radiation on hematopoiesis, the radiation group were given a four Gy X-ray total-body irradiation. The hematological parameters were analyzed at different time points. Based on the results, the level of peripheral platelets gradually decreased from days 4 to 7 and slightly recovered on day 11 after irradiation (Figure 1A). The white blood cells (WBCs) were more sensitive to irradiation than platelets (Figure 1A,B). The peripheral WBC count decreased immediately after irradiation and remained at a low level from day 0 to 11 (Figure 1B). The number of peripheral RBC also decreased on days 4, 7, and 11 after exposure to irradiation (Figure 1C). Besides the direct harm of irradiation on platelets, the decrease in platelet counts may also be caused by insufficient production of megakaryocytes. H&E staining revealed that the number and size of BM and spleen megakaryocytes in the irradiation group were markedly lower than those in the normal group (Figure 1D–G). Such finding suggests that irradiation inhibits the production and maturation of megakaryocytes in the BM and spleen. The expression levels of the megakaryocytic lineage-specific differentiation marker CD41 and maturation marker CD42d were further detected by flow cytometry. Based on the results, the expression levels of CD41 and CD42d in BM cells were significantly reduced after irradiation (Figure 1H,I), suggesting that irradiation inhibited megakaryocyte differentiation. Moreover, the effect of irradiation on apoptosis was assessed by Annexin V-FITC/PI staining. The results showed that irradiation markedly promoted the apoptosis of BM cells (Figure 1J,K). Taken together, these results suggest that irradiation can cause myelosuppression, inhibit megakaryocyte differentiation, and promote the apoptosis of BM cells.

### 2.2. Multivariate Statistical Analysis of the Metabolomics Research

A principal component analysis (PCA) was used for the initial grouping of normal and irradiated mice to derive more intuitive and visual results of the sample analysis. The sample distribution areas of the normal and model groups were basically separated, and all samples fell within the 95% confidence interval, which indicates that the serum metabolites were significantly changed in irradiated mice (Figure 2A). Furthermore, the use of OPLS-DA can better reflect the differences in serum metabolites between normal and irradiated mice, maximizing the separation and mining different metabolites. As shown in Figure 2B, the OPLS-DA score scatter plots showed that the samples of model mice and normal mice were completely separated and fell into different regions, indicating that the metabolism of the model group was different from that of the normal group. The R^2^Y value of OPLS-DA models for LC-MS (ESI^+^) was 0.856 and the Q^2^ value was 0.751, the fitness and predictive ability of the model were evaluated using R^2^ and Q^2^, and 100 permutation tests were performed on R^2^ and Q^2^ as shown in Figure 2C, which suggests good fitness and predictive ability of the model. Finally, the loading S-plot was initially screened for metabolite ions that differed significantly between normal and irradiated mice (Figure 2D). Thus, the results of the multivariate statistical analysis showed significant differences in the serum metabolic profiles between normal and irradiated mice, indicating that specific metabolites were altered in mice after radiation exposure.

### 2.3. Identification of Potential Biomarkers of Thrombocytopenia

The variable importance in projection (VIP) parameters in the OPLS-DA model and *P*-value in the Student’s *t*-test were used to screen differential variables, which were uploaded to the One-Map. The matching results were integrated with HMDB metabolite database search and KEGG database search to identify differential metabolites. As shown in Table 1, 66 different metabolites were identified and tentatively regarded as the potential biomarkers of thrombocytopenia, of which 56 were up-regulated and 10 were down-regulated in irradiated mice compared to normal mice on day 11 after irradiation. A heatmap was drawn based on the relative concentrations of the metabolites in each sample, which reflected the metabolite trends (Figure 3). L-tryptophan, LysoPC (17:0), and D-Sphinganine were also mapped to important pathways; therefore, the relative contents of the three metabolites on day 11 were analyzed between normal and irradiated mice, as shown in Figure 4. Compared to normal mice, the levels of the three metabolites increased one to three times in irradiated mice. On day 30 after irradiation, the platelet counts in irradiated mice increased to normal levels, and the metabolite contents at this time point were detected using metabolomics. The result showed that there was no significant difference in the content of L-tryptophan, LysoPC (17:0), and D-Sphinganine between irradiated mice and normal mice after 30 days of recovery from radiation exposure (*p* > 0.05) (Figure 5). These results indicate that the contents of L-tryptophan, LysoPC (17:0), and D-Sphinganine are closely related to platelet levels.

### 2.4. Biological Pathway Analysis of Radiation-Induced Thrombocytopenia

A pathway analysis was performed to discover the biological metabolic pathways that may be associated with radiation-induced thrombocytopenia. The data revealed 16 metabolic pathways that were associated with radiation injury (Figure 6, Table 2). The metabolic pathways with impact > 0.1 contained five pathways, including glycerophospholipid metabolism, nicotinate and nicotinamide metabolism, sphingolipid metabolism, inositol phosphate metabolism, and tryptophan metabolism. Significant fluctuations in different metabolites caused by radiation can lead to metabolic disorders, which may be one of the causes of radiation-induced thrombocytopenia in mice. In conclusion, disorders in glycerophospholipid metabolism, nicotinate and nicotinamide metabolism, sphingolipid metabolism, inositol phosphate metabolism, and tryptophan metabolism are involved in radiation-induced thrombocytopenia.

### 2.5. Effects of Metabolites on Megakaryocyte Differentiation and the Apoptosis of K562 Cells

As the suppression of megakaryocyte differentiation and the promotion of BM cell apoptosis caused by radiation might be mediated by metabolites, the influence of key metabolites on the human erythroleukemia cell line K562 were evaluated. As expected, we found that L-tryptophan significantly inhibited megakaryocyte differentiation of K562 cells in a concentration-dependent manner (Figure 7A,B). L-tryptophan, LysoPC (17:0), and D-sphinganine remarkably promoted apoptosis in a concentration-dependent manner (Figure 7C,D). Collectively, these results indicate that L-tryptophan, LysoPC (17:0), and D-sphinganine are the main metabolites that mediate radiation-induced thrombocytopenia via the inhibition of megakaryocyte differentiation and promotion of megakaryocyte apoptosis.

## 3. Discussion

Radiation exposure exists in all aspects of life. In addition to accidental exposure such as nuclear leakage accidents, radiation treatment, such as nuclear medicine, is the main source of radiation exposure. Despite advances in radiotherapy in recent years, damage to healthy tissues, especially the hematopoietic system, remains an inevitable problem [35]. Thrombocytopenia is a primary complication of radiotherapy that can be life threatening. Platelet count was reported to be an important indicator of surviving radiation exposure [36,37]. However, the clinical treatment of thrombocytopenia is usually insufficient. Therefore, the pathogenesis of thrombocytopenia must be understood more deeply and potential therapeutic targets or intervention pathways must be discovered for the treatment of thrombocytopenia.

Megakaryocytes in the BM produce platelets through proliferation, differentiation, migration, maturation, and platelet release. Platelet count is directly determined by the number and quality of megakaryocytes [38]. Damage to the hematopoietic system induced by ionizing radiation causes damage to hematopoietic cells [39]. Our results showed that platelet levels in mice began to decrease on day 4 after radiation exposure and reached their lowest levels on day 7. Further, the number of megakaryocytes was significantly reduced in the BM and the spleen of irradiated mice. Radiation decreases the number of HSCs and their ability to self-renew and differentiate, which might be caused by ROS [40]. Consistent with previous studies, our study revealed that radiation-induced thrombocytopenia was multifaceted. Radiation not only directly damages mature platelets through ROS, but also reduces megakaryocyte formation in the BM and spleen and inhibits megakaryocyte differentiation into platelets.

Untargeted metabolomics is helpful for conducting comprehensive and systematic analysis of metabolic profiles and identifying differential metabolites, which reveals the direct effects of radiation. Significant changes in plasma metabolites have been demonstrated to occur in rats after radiation exposure, accompanied by a significant radiation dose–effect relationship [32]. According to our findings, radiation could cause changes in the metabolic profiles of mice. Further, PCA intuitively revealed the changes via dimensionality reduction of the data. OPLS-DA could find the variables most related to grouping factors, so the metabolic profiles of normal and irradiated mice could be better separated by OPLS-DA. In the OPLS-DA model, the serum metabolic profiles of normal and irradiated mice were significantly separated, suggesting changes in the levels of specific metabolites in the serum of mice after radiation exposure. Sixty-six different metabolites that may be closely related to radiation-induced thrombocytopenia were identified. Further, multiple pathways were found to be associated with radiation-induced thrombocytopenia, most of which have been reported to be associated with radiation injury or thrombocytopenia [25,41,42]. However, the upstream and downstream metabolites involved in these pathways are different and diverse, so the specific metabolites that mediate radiation injury need to be further investigated. We combined differential metabolite and pathway analysis to screen out three metabolites, D-sphinganine, LysoPC (17:0), and L-Tryptophan, which were up-regulated by radiation. What’s more, when the platelet counts recovered to normal levels in mice at 30 days after radiation exposure, no differences in the content of the three metabolites were found in irradiated mice compared to normal mice. Therefore, it is reasonable to assume that D-sphinganine, LysoPC (17:0), and L-Tryptophan are potential biomarkers for radiation-induced thrombocytopenia.

Glycerophospholipids are among the most abundant and complex phospholipids in the body. Glycerophospholipids are not only involved in the formation of biological membranes but are also components of bile and membrane-surface active substances. In addition, glycerophospholipids participate in protein recognition and signal transduction through the cell membrane to maintain normal energy and various metabolisms in the body [43]. Therefore, glycerophospholipid metabolism is one of the most reported disordered pathways in metabolomics research [44,45]. LysoPC (17:0) is a type of phosphatidylcholine, which is not only one of the small molecules involved in glyceropholipid metabolism, but is also a major component of the lipid bilayer structure of cell membranes and lipoproteins that maintain blood cell integrity and normal morphology. Glycerophospholipid metabolism abnormalities may be related to phospholipase, which catalyzes the ester bond breakage of glycerophospholipids to produce lysophospholipids. Lysophospholipids are a class of surface-active substances that can rupture RBC and other cell membranes, and cause hemolysis or cell necrosis [46]. Our results showed a greater than three-fold increase in LysoPC (17:0) in irradiated mice compared to normal mice, and LysoPC (17:0) induced apoptosis of K562 cells. Therefore, we hypothesized that radiation increases the relative contents of lysophospholipids, including LysoPC (17:0), which causes thrombocytopenia and may play a role in two ways. One, lysophospholipids act directly on platelets causing their cell membranes to be destroyed by rupture. Two, lysophospholipids reduce platelet formation by inducing the apoptosis of megakaryocytes and other hematopoietic cells.

Sphingolipids are considered an integral part of the cell structure and important signaling molecules, which are closely related to cell growth, aging, meiosis, maturation, and death [47]. Sphingolipid metabolism has a significant regulatory effect on human HSCs [48]. Tadbir et al. suggested that the lack of functional 3-ketodihydrosphingosine reductase causes sphingolipid dysregulation, which impedes proplatelet formation and causes thrombocytopenia [42]. Ionizing radiation has been reported to cause various disorders in sphingolipid metabolism, and its downstream metabolite, ceramide, has been most frequently studied [49,50]. Radiation-induced apoptosis is directly related to ceramide accumulation, which might be directly involved in the molecular mechanism of radiation injury to cells [51]. Here, our experiments identified an upstream metabolite of sphingolipid metabolism, D-sphinganine, which may directly participate in radiation-induced cellular damage. Sphingolipids and their derivatives, D-sphinganine, are potent growth inhibitors, and exogenous sphingolipids can induce cycle arrest and the apoptosis of many cell lines [52]. The content of D-sphinganine doubled in irradiated mice compared to that in normal mice. D-sphinganine induced the apoptosis of K562 cells. Therefore, we speculate that radiation induces an abnormal increase in D-sphinganine levels by disrupting sphingolipid metabolism, which induces the apoptosis of hematopoietic cells and damages hematopoietic function in mice.

L-tryptophan, an essential amino acid in humans and animals, participates in various physiological processes, including neuronal function, gut homeostasis, and protein biosynthesis [53]. Tryptophan metabolism is closely associated with the occurrence and development of numerous clinical diseases. According to recent studies, the pathogenesis of immune thrombocytopenia is related to abnormal tryptophan metabolism mediated by indoleamine 2,3-dioxygenase (IDO), which decreases IDO activity and increases tryptophan concentration [54]. Our study suggested that the tryptophan metabolism was abnormal and the L-tryptophan content increased after X-ray irradiation. However, whether L-tryptophan is directly involved in radiation-induced cell injury is unclear. In conducting in vitro experiments, we found that L-tryptophan not only induced apoptosis, but also inhibited the differentiation of K562 cells. Therefore, tryptophan may directly cause radiation-induced cell injury by inducing apoptosis and inhibiting hematopoietic cells differentiation, ultimately leading to hematopoietic disorders.

In our study, the mechanism of action of radiation-induced thrombocytopenia was found to involve multiple pathways. First, radiation can directly damage platelets, which causes a remarkable decline in platelet count. Second, radiation reduces platelet formation by inducing the apoptosis of hematopoietic cells and inhibiting the differentiation of megakaryocytes. Our studies suggest that multiple endogenous metabolites are involved in this process. The causal relationship between abnormal metabolites and cell malfunction is complex and difficult to completely elucidate. Except for our hypothesis, there could be the other possibility that irradiation affects metabolism by affecting the cells. We speculated that cells and metabolites can interact with each other when radiation occurs. In future studies, we will determine the direct effects of endogenous metabolites on hematopoiesis and validate their functions in vitro and in vivo.

## 4. Materials and Methods

### 4.1. Chemicals and Reagents

Acetonitrile and formic acid were of LC-MS grade and were purchased from Thermo Fisher (Waltham, MA, USA). Ultrapure water was obtained using a Milli-Q water purification system (Billerica, MA, USA). 1-heptadecanoyl-2-hydroxy-sn-glycero-3-phosphocholine [LysoPC (17:0)] (purity 96%, as determined by qNMR) was purchased from Macklin (Shanghai, China). L-Tryptophan (purity = 99.96%, as determined by HPLC) was obtained from Bidepharm (Shanghai, China). Dihydrosphingosine (purity 99.99%, as determined by qNMR) was obtained from Toronto Research Chemicals (Toronto, ON, Canada).

### 4.2. Animals and Treatment

The Kunming (KM) mice, 2 months old and 18–22 g in weight, were purchased from Da-suo Bio-technology Co., Ltd. (Chengdu, Sichuan, China). The mice were kept under a specific pathogen-free room with standard conditions (25 ± 2 °C, 55 ± 10% humidity and 12 h light–dark cycle) and administered standard food and water ad libitum. Mice were acclimatized for 7 days before radiation. All experimental procedures were reviewed and approved by the laboratory animal ethics committee of the Southwest Medical University (No. 20211123-014). Fifty-two KM mice were randomly divided into 2 groups (13 males and 13 females in each group): a normal group and a radiation group. Except for mice in the normal group, all mice were given a 4 Gy X-ray total body irradiation to induce myelosuppression. Thirty-two mice (16 normal mice and 16 irradiated mice) were sacrificed after 11 days of the experiment and serum samples were collected for untargeted metabolomics analysis. The remaining 20 mice were sacrificed on day 30 of the experiment and their serum was collected to detect the relative content of differential metabolites.

### 4.3. Hematologic Parameter Analysis

A total of 40 μL peripheral blood was obtained from the fundus vein plexus on day 11. After treatment with 160 μL of diluent, the hematological parameters were detected by the automatic blood cell analyzer (Sysmex XT-1800i/2000IV, Kobe, Japan).

### 4.4. Histology Analysis

Eleven days after irradiation, 3 mice were randomly selected from each group and their femurs were moved out. The femurs were stored in 10% formaldehyde for 1 day and decalcified with a decalcifying solution for more than 1 month. Then, the femurs were embedded in paraffin and cut into 5 mm sections. The samples were stained with hematoxylin and eosin (H&E) and taken photos under an Olympus BX51 microscope (Olympus Optical, Shinjuku, Tokyo, Japan).

### 4.5. Flow Cytometry Analysis of BM Megakaryocyte Differentiation

Total BM cells were flushed out from femurs by saline solution and treated with RBC lysis buffer (Beijing 4A Biotech, Beijing, China) to remove red blood cells (RBCs). The samples were incubated with FITC-conjugated anti-CD41 (BioLegend, San Diego, CA, USA) and PE-conjugated anti-CD42d (BioLegend, San Diego, CA, USA) for 30 min on ice in the dark. Flow cytometry were carried out by BD FACSCanto II flow cytometer (BD Biosciences, San Jose, CA, USA).

### 4.6. Flow Cytometry Analysis of BM Cell Apoptosis

After treatment with RBC lysis buffer (Beijing 4A Biotech, Beijing, China), the BM cells were incubated with Annexin V-FITC and PI (BD Biosciences, San Jose, CA, USA) for 15 min at 25 °C in the dark. Then the cell apoptosis was measured by the BD FACSCanto II flow cytometer (BD Biosciences, San Jose, CA, USA).

### 4.7. Sample Preparation

Ice-cold methanol (800 μL) was added to 200 μL serum sample and mixed by vortex for 1 min. The mixture was then left for 10 min. After centrifugation at 4 °C for 15 min at 10,000 rpm, the supernatant was transferred into a new centrifuge tube and freeze-dried. The residual powder was redissolved in 200 μL methanol–water (20:80, v:v) and the supernatant was transferred to a sample vial for UHPLC–QTOF MS analysis. Quality control (QC) samples were obtained by mixing serum (10 μL) from each sample. During data acquisition, QC samples were injected every 5 samples to monitor system stability. All samples were tested in the order of the normal and model groups.

### 4.8. Untargeted Metabolomics Analysis

Metabolomics profiling of serum samples in the normal and model groups were performed on UHPLC (Exion)—QTOF (X500R) MS system (SCIEX, Framingham, MA, USA) for untargeted metabolomics in positive mode. Chromatographic separation was conducted by an Exion UHPLC system coupled to a C_18_ column (Kinetex C_18_, 2.1 × 100 mm, 2.6 μm). The analytical system was set as follows: column temperature 40 °C, flow rate 0.30 mL min^−1^, and the injection volume 20 µL. The mobile phase was 0.1% formic acid–water (v:v) and acetonitrile (B) with the following gradient program: 0–1 min, 5% B; 1–14 min, 5-100% B; 14–15 min, 100% B; 15–15.01 min, 100–5% B; 15.01–20 min, 5% B.

A X500R QTOF MS system was equipped with a Turbo V^TM^ source with Twin Sprayer probe for electrospray ionization (ESI). The MS parameters were set as follows: spray temperature: 500 °C; full scan mass range, *m/z* 100–1500 Da; ion source gas 1 and 2: 50 psi; curtain gas: 35 psi; CAD gas: 7 psi. Information-dependent acquisition (IDA) mode was used for the automated MS/MS data acquisition. The specific parameters with dynamic background subtraction were as follows: maximum candidate ions: 10; intensity threshold exceeds: 400 cps; exclude former candidate ions for 5 s after 1 occurrence.

### 4.9. Data Processing and Multivariate Analysis

The LC-MS original datasets were processed by One-MAP/PTO software (Dalian Chem Date Solution Technology Co. Ltd., Dalian, Liaoning, China), and retention time correction, baseline filtration, peak detection, and alignment were performed. The data processing software with centWave mode was set as follows: ppm: 20; minimum peak width: 5; maximum peak width: 5; prefilterk: 4; prefilterl: 10; bw1:10; bw2:7; method: obiwarp; SNthresh: 5; mzdiff: −0.001. The retention time and *m/z* data for each peak were obtained. Missing values of more than 50% for some metabolic characteristics were deleted, and the missing values for the remaining variables were filled with minimum values. Synchronously, some variables were filtered by the 80% rule, in which the proportion of variables with zero value in any sample group was greater than 20%. The filtered and aligned data matrix was obtained and normalized using total area normalization.

The peak data matrix was introduced into SIMCA 14.1 software (Umetrics, Sweden) for multivariate analysis including principal component analysis (PCA) and orthogonal partial least-squares-discriminant analysis (OPLS-DA). The OPLS-DA model was validated using rigorous methods [55]. Two-tailed Student’s *t*-tests were performed via SPSS 19.0 software (SPSS Inc., Chicago, IL, USA). Different endogenous metabolites with variable importance in projection (VIP) values > 1.0 in OPLS-DA and two-tailed Student’s *t*-test *p* values < 0.05 were considered potential discriminant metabolites. Metabolites were identified using the One-Map platform (http://www.5omics.com/, accessed on 1 June 2021), which integrates the Kyoto Encyclopedia of Genes and Genomes (KEGG; http://www.kegg.jp, accessed on 1 June 2021), Human Metabolome Database (HMDB; http://www.hmdb.ca, accessed on 1 June 2021), and a self-built standard database, etc. Pathway analysis was performed using MetaboAnalyst 5.0 (https://www.metaboanalyst.ca/, accessed on 1 June 2021) and pathways with impact values > 0.1 were deemed to be potentially related to hematopoietic system injury.

### 4.10. Analysis of Megakaryocyte Differentiation

The K562 cells were seeded into 6-well plate at a density of 2 × 10^4^ cells mL^−1^ and co-treated with phorbol 12-myristate 13-acetate (1 μM) (Sigma, St. Louis, MO, UAS) and different concentrations of metabolites for 6 days. Then the cells were incubated with FITC conjugated anti-CD41 (Biolegend, San Diego, CA, USA) and PE conjugated anti-CD42b (Biolegend, San Diego, CA, USA) and analyzed by the BD FACSCanto II flow cytometer (BD Biosciences, San Jose, CA, USA).

### 4.11. Cell Apoptosis Assay

After different treatment, the cell apoptosis was measured using BD Pharmingen™ FITC Annexin V Apoptosis Detection Kit I (BD Biosciences, San Jose, CA, USA) according to the manufacturer’s instructions.

### 4.12. Statistical Analysis

Data are expressed as mean ± SD and all data were statistically analyzed by SPSS 19.0 software (SPSS Inc., Chicago, IL, USA). *p* values were considered statistically significant at *p* < 0.05.

## 5. Conclusions

The present study is novel as it explores the potential pathogenesis and pathways involved in radiation-induced thrombocytopenia based on UHPLC/MS-based serum metabolomics and validates the effects of key metabolites on the apoptosis and differentiation of cells. This study also aids in the screening for potential biomarkers of radiation-induced thrombocytopenia. Radiation can not only directly damage platelets, but also induce hematopoietic cell apoptosis and inhibit megakaryocyte differentiation to reduce platelet formation, leading to thrombocytopenia. A multivariate statistical analysis showed significant changes in the serum metabolic profile of irradiated mice; 66 differential metabolites were identified, 56 of which were up-regulated and 10 were down-regulated in irradiated mice compared to normal mice. A pathway analysis showed that radiation-induced thrombocytopenia was associated with disturbances in glycerophospholipid metabolism, nicotinate and nicotinamide metabolism, sphingolipid metabolism, inositol phosphate metabolism, and tryptophan metabolism. Three specific metabolites, D-sphinganine, LysoPC (17:0), and L-tryptophan belongs to sphingolipid metabolism, glycerophospholipid metabolism, and tryptophan metabolism, respectively. During the radiation-induced thrombocytopenia phase, the contents of these three metabolites were abnormally elevated in mice. As platelet levels recovered in irradiated mice, the contents of D-sphinganine, LysoPC (17:0), and L-tryptophan were restored to normal levels. The results of cellular assays showed that these metabolites significantly induced the apoptosis of K562 cells and L-tryptophan inhibited K562 cell differentiation. The three specific metabolites were highly correlated with the cellular regulation of radiation injury, which can serve as potential biomarkers for radiation-induced thrombocytopenia. Intervention based on the dynamic changes in different metabolites and related metabolic pathways may be a promising strategy for the treatment of radiation-induced thrombocytopenia, which requires further study. We will further explore the dynamic changes in metabolites and validate their function in the regulation of megakaryocyte differentiation in vivo and in vitro.

## Figures and Tables

**Figure 1 ijms-23-07978-f001:**
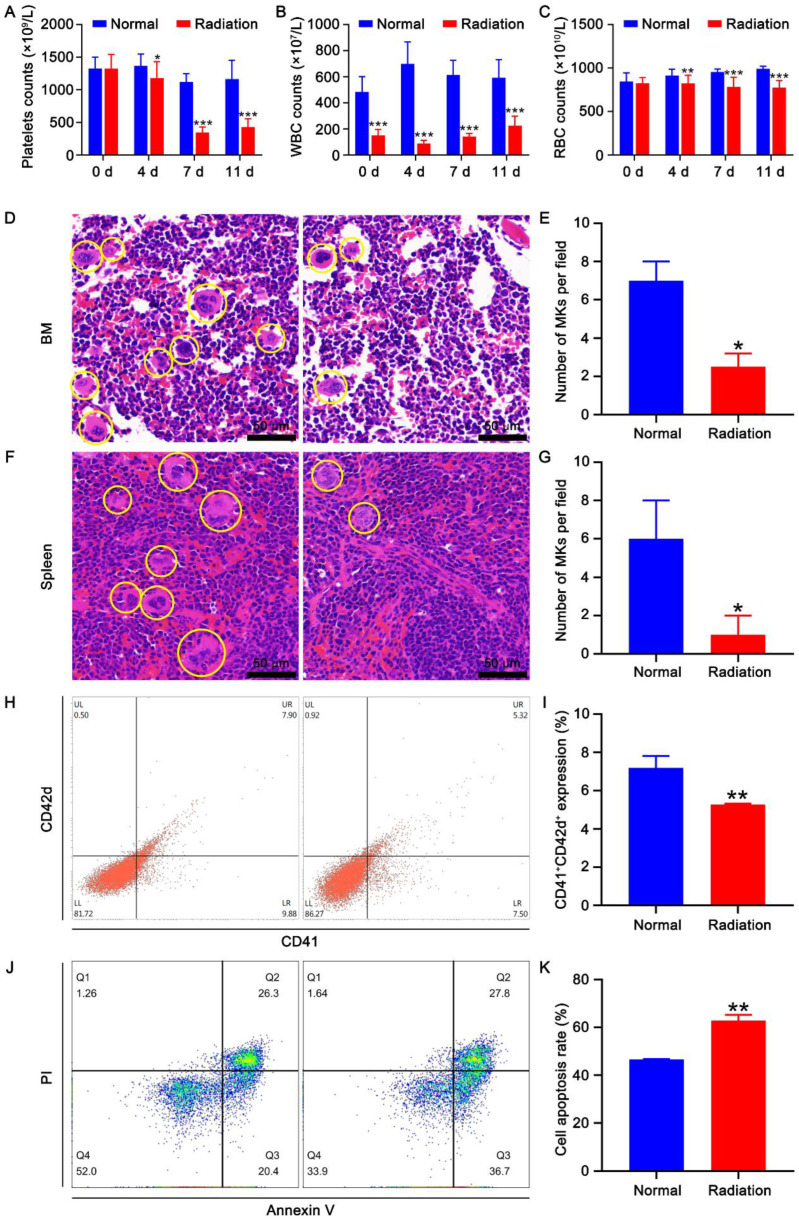
Effect of irradiation on hematopoiesis. Peripheral platelet (**A**), WBC (**B**), and RBC (**C**) counts in mice at different time points after irradiation. Thirty-two mice divided into two groups (eight males and eight females in each group) were used. Data represent the mean ± SD of three independent experiments. * *p* < 0.05, ** *p* < 0.01, *** *p* < 0.001 vs. the normal group. H&E staining of BM (**D**) and spleen (**F**) in normal and irradiated mice. The histograms represent the number of megakaryocytes in the BM (**E**) and spleen (**G**) of normal and irradiated mice. Data represent the mean ± SD of three independent experiments. * *p* < 0.05 vs. the normal group. (**H**) Flow cytometry analysis of the expression of CD41 and CD42d in the BM cells of normal and irradiated mice. (**I**) The histogram illustrating the proportion of CD41 + CD42d + cells in each group. Data represent the mean ± SD of three independent experiments. ** *p* < 0.01 vs. the normal group. (**J**) Apoptosis analysis of the BM cells of normal and irradiated mice. (**K**) The histogram depicting the apoptosis rate of BM cells in each group. Data represent the mean ± SD of three independent experiments. ** *p* < 0.01 vs. the normal group.

**Figure 2 ijms-23-07978-f002:**
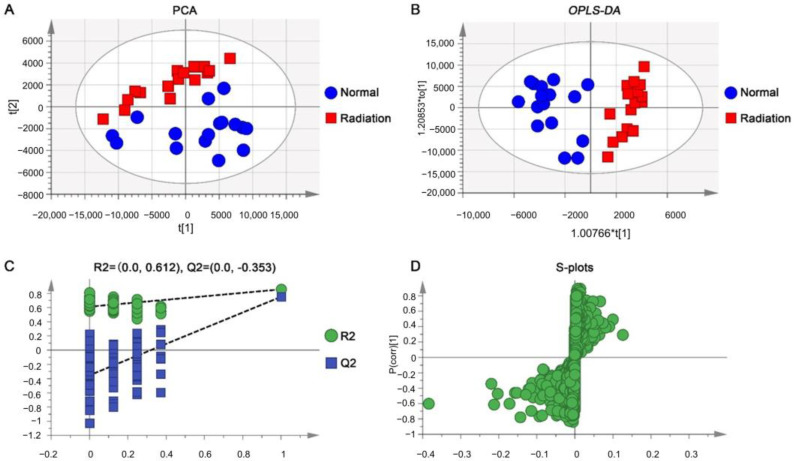
Radiation-induced metabolomics phenotypes of mice in the ESI+ model. PCA score plot (**A**) and OPLS-DA score plot (**B**) for serum metabolic profiling of normal and irradiated mice, permutation validation (*n* = 100) (**C**), and S-plots (**D**) between normal and irradiated mice. * ESI+: Electrospray ionization source in positive-ion mode.

**Figure 3 ijms-23-07978-f003:**
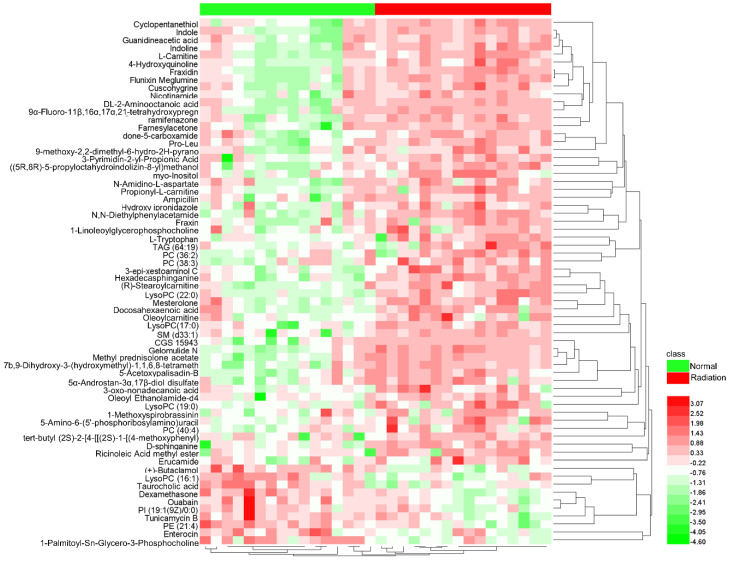
Heatmap of 66 differential metabolites in mice serum samples. The color is proportional to the intensity of change in metabolites. Green indicates down-regulation and red indicates up-regulation.

**Figure 4 ijms-23-07978-f004:**
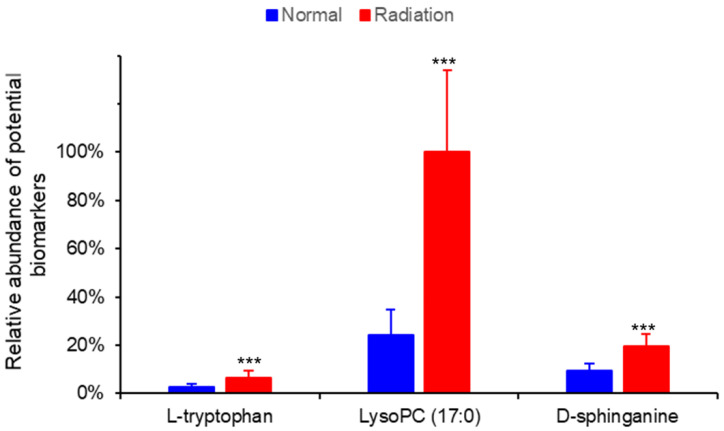
Relative abundance of three potential biomarkers in normal and irradiated mice. Data represent mean ± SD, *** *p* < 0.001 vs. normal mice.

**Figure 5 ijms-23-07978-f005:**
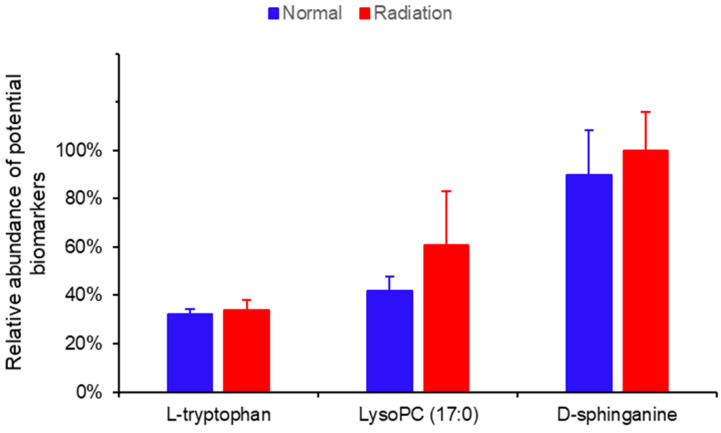
Relative abundance of three potential biomarkers in the extract ion chromatogram of normal and irradiated mice. Data represent mean ± SD.

**Figure 6 ijms-23-07978-f006:**
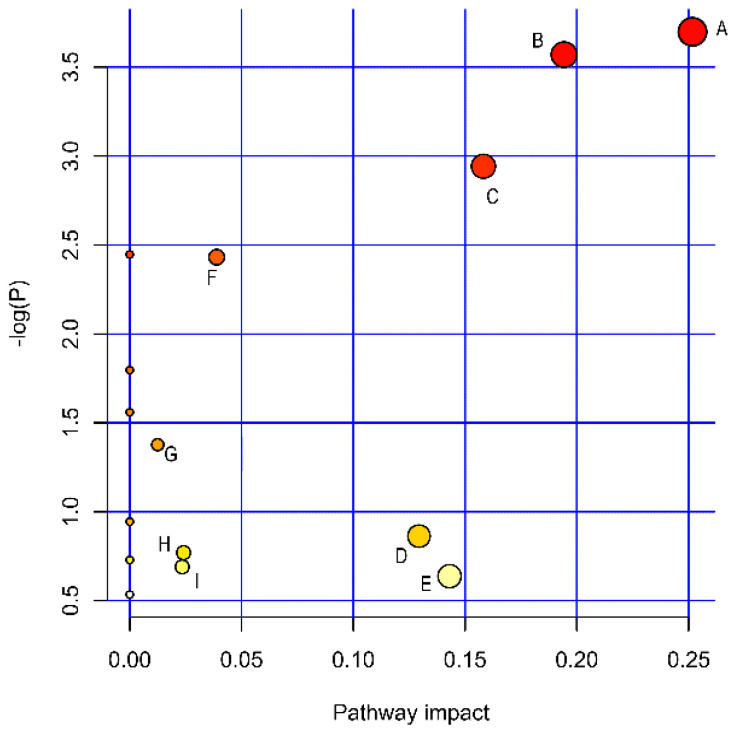
Pathway analysis of radiation-induced thrombocytopenia. Glycerophospholipid metabolism (**A**), nicotinate and nicotinamide metabolism (**B**), sphingolipid metabolism (**C**), inositol phosphate metabolism (**D**), tryptophan metabolism (**E**), Phosphatidylinositol signaling system (**F**), Glycerolipid metabolism (**G**), Glycine, serine, and threonine metabolism (**H**), and Arginine and proline metabolism (I).

**Figure 7 ijms-23-07978-f007:**
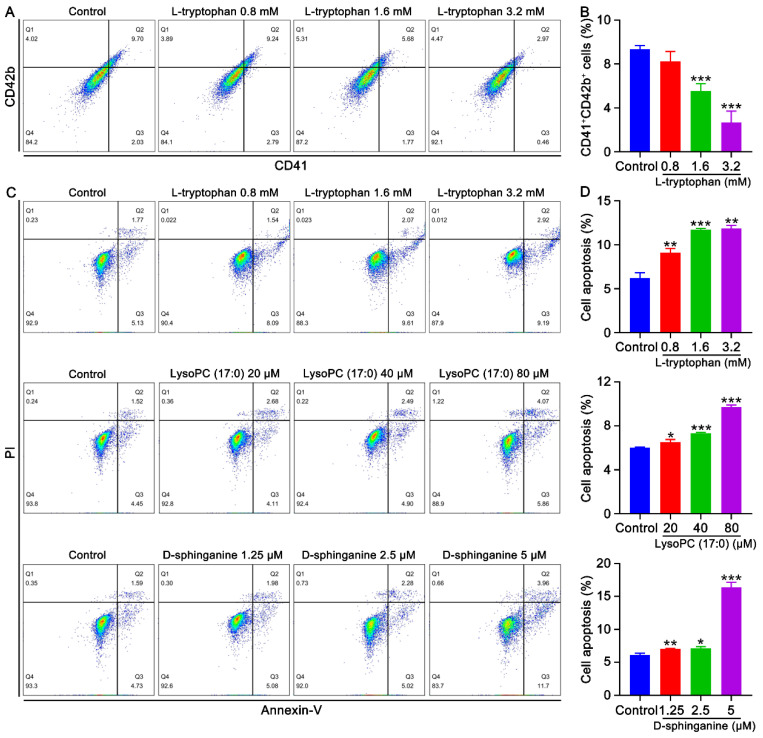
Effects of metabolites on the differentiation and apoptosis of K562 cells. (**A**) Flow cytometry analysis of the expression of CD41 and CD42b after the treatment of K562 cells with different concentrations (0.8, 1.6, and 3.2 mM) of L-tryptophan for 6 days. (**B**) The histogram represents the proportion of CD41 + CD42b + cells in the control and L-tryptophan-treated groups. Data represent the mean ± SD of three independent experiments. *** *p* < 0.001 vs. the control group. (**C**) Apoptosis analysis of K562 cells treated with different concentrations of L-tryptophan (0.8, 1.6, and 3.2 mM), LysoPC (17:0) (20, 40, and 80 μM), and D-sphinganine (1.25, 2.5 and 5 μM) for 6 days. (**D**) The histogram represents the apoptosis rate of each group. Data represent the mean ± SD of three independent experiments. * *p* < 0.05, ** *p* < 0.01, *** *p* < 0.001 vs. the control group.

**Table 1 ijms-23-07978-t001:** Results of metabolite identification.

No.	ESI Mode	Metabolites	Chemical Formula	RT (min)	m/z	Ion Type	VIP Value	Normal vs. Radiation Trend
1	Pos	Cyclopentanethiol	C_5_H_10_S	2.184	103.0539	M + H	1.8754	↑
2	Pos	Indole	C_8_H_7_N	3.771	118.0647	M + H	1.8679	↑
3	Pos	Guanidineacetic acid	C_3_H_7_N_3_O_2_	2.197	118.0648	M + H	1.5384	↑
4	Pos	Indoline	C_8_H_9_N	2.195	120.0804	M + H	1.9493	↑
5	Pos	Nicotinamide	C_6_H_6_N_2_O	1.256	123.0552	M + H	1.4577	↑
6	Pos	4-Hydroxyquinoline	C_9_H_7_NO	4.695	146.0598	M + H	1.8679	↑
7	Pos	N1-Methyl-2-pyridone-5-carboxamide	C_7_H_8_N_2_O_2_	1.686	153.0657	M + H	1.685	↑
8	Pos	3-Pyrimidin-2-yl-Propionic Acid	C_7_H_8_N_2_O_2_	2.621	153.0657	M + H	1.5834	↑
9	Pos	DL-2-Aminooctanoic acid	C_8_H_17_NO_2_	1.670	160.1330	M + H	1.8437	↑
10	Pos	L-Carnitine	C_7_H_15_NO_3_	1.581	162.1123	M + H	1.8223	↑
11	Pos	N-Amidino-L-aspartate	C_5_H_9_N_3_O_4_	1.398	176.0657	M + H	1.2443	↑
12	Pos	Hydroxy ipronidazole	C_7_H_11_N_3_O_3_	4.172	186.0911	M + H	1.298	↑
13	Pos	N,N-Diethylphenylacetamide	C_12_H_17_NO	7.316	192.1382	M + H	1.973	↑
14	Pos	((5R,8R)-5-propyloctahydroindolizin-8-yl)methanol	C_12_H_23_NO	0.224	198.1850	M + H	1.6674	↑
15	Pos	myo-Inositol	C_6_H_12_O_6_	1.046	203.0526	M + Na	1.4349	↑
16	Pos	L-Tryptophan	C_11_H_12_N_2_O_2_	4.775	205.0970	M + H	1.3927	↑
17	Pos	Propionyl-L-carnitine	C_10_H_19_NO_4_	1.730	218.1384	M + H	1.3618	↑
18	Pos	Fraxidin	C_11_H_10_O_5_	0.257	223.0635	M + H	2.1555	↑
19	Pos	Cuscohygrine	C_13_H_24_N_2_O	7.730	225.1957	M + H	1.7674	↑
20	Pos	Pro-Leu	C_11_H_20_N_2_O_3_	1.697	229.1545	M + H	1.8095	↑
21	Pos	3-epi-xestoaminol C	C_14_H_31_NO	8.755	230.2475	M + H	2.1067	↑
22	Pos	ramifenazone	C_14_H_19_N_3_O	4.372	246.1697	M + H	1.7439	↑
23	pos	9-methoxy-2,2-dimethyl-6-hydro-2H-pyrano [5,6-c] quinolin-5-one	C_15_H_15_NO_3_	1.176	258.1104	M + H	1.2531	↑
24	Pos	Farnesylacetone	C_18_H_30_O	12.776	263.2360	M + H	1.6348	↑
25	Pos	Hexadecasphinganine	C_16_H_35_NO_2_	8.773	274.2734	M + H	2.1172	↑
26	Pos	1-Methoxyspirobrassinin	C_12_H_12_N_2_O_2_S_2_	0.293	281.0506	M + H	1.3219	↑
27	Pos	CGS 15943	C_13_H_8_ClN_5_O	8.997	286.0563	M + H	1.8927	↑
28	Pos	Flunixin Meglumine	C_14_H_11_F_3_N_2_O_2_	0.366	297.0824	M + H	2.0744	↑
29	Pos	D-sphinganine	C_18_H_39_NO_2_	9.670	302.3046	M + H	1.8162	↑
30	Pos	Mesterolone	C_20_H_32_O_2_	12.639	305.2465	M + H	1.9037	↑
31	Pos	3-oxo-nonadecanoic acid	C_19_H_36_O_3_	11.835	313.2724	M + H	1.4141	↑
32	Pos	Ricinoleic Acid methyl ester	C_19_H_36_O_3_	12.266	313.2727	M + H	1.3506	↑
33	Pos	Docosahexaenoic acid	C_22_H_32_O_2_	12.614	329.2464	M + H	1.6971	↑
34	Pos	Oleoyl Ethanolamide-d4	C_20_H_35_D_4_NO_2_	10.608	330.3354	M + H	1.2375	↑
35	Pos	Erucamide	C_22_H_43_NO	13.608	338.3402	M + H	1.2556	↑
36	Pos	Ampicillin	C_16_H_19_N_3_O_4_S	3.933	350.1235	M + H	1.2683	↑
37	Pos	5-Amino-6-(5’-phosphoribosylamino)uracil	C_9_H_15_N_4_O_9_P	0.397	355.0696	M + H	1.2117	↑
38	Pos	(+)-Butaclamol	C_25_H_31_NO	5.351	362.2413	M + H	1.7738	↓
39	Pos	Fraxin	C_16_H_18_O_10_	0.284	371.1013	M + H	1.4584	↑
40	Pos	Dexamethasone	C_22_H_29_FO_5_	5.407	393.2098	M + H	1.9392	↓
41	Pos	9α-Fluoro-11β,16α,17α,21- tetrahydroxypregn-4-ene-3,20-dione	C_21_H_29_FO_6_	9.033	397.2015	M + H	1.9768	↑
42	Pos	Gelomulide N	C_24_H_32_O_7_	9.038	415.2110	M + H−H_2_O	2.1011	↑
43	Pos	Methyl prednisolone acetate	C_24_H_32_O_6_	9.035	417.2177	M + H	2.0157	↑
44	Pos	Oleoylcarnitine	C_25_H_47_NO_4_	11.324	426.3560	M + H	1.3862	↑
45	Pos	(R)-Stearoylcarnitine	C_25_H_49_NO_4_	11.934	428.3721	M + H	2.0445	↑
46	Pos	7b,9-Dihydroxy-3-(hydroxymethyl)-1,1,6,8-tetramethyl-5-oxo-1,1a,1b,4,4a,5,7a,7b,8,9-decahydro-9aH-cyclopropa [3,4] benzo [1,2-e] azulen-9a-ylacetate	C_22_H_30_O_6_	9.038	432.2376	M + CAN + H	1.9373	↑
47	Pos	5-Acetoxypalisadin-B	C_17_H_26_Br_2_O_3_	9.030	437.1932	M + H	1.7467	↑
48	Pos	5α-Androstan-3α,17β-diol disulfate	C_19_H_32_O_8_S_2_	9.007	453.1678	M + H	1.3525	↓
49	Pos	Enterocin	C_22_H_20_O_10_	0.865	467.1025	M + Na	1.2918	↑
50	Pos	LysoPC (16:1)	C_24_H_48_NO_7_P	8.832	494.3229	M + H	1.3253	↓
51	Pos	1-Palmitoyl-Sn-Glycero-3-Phosphocholine	C_24_H_50_NO_7_P	9.853	496.3374	M + H	1.6346	↓
52	Pos	LysoPC(17:0)	C_25_H_52_NO_7_P	10.660	510.3535	M + H	1.8249	↑
53	Pos	Taurocholic acid	C_26_H_45_NO_7_S	8.821	516.3061	M + H	1.6262	↓
54	Pos	1-Linoleoylglycerophosphocholine	C_26_H_51_NO_7_P	9.991	520.3378	M + H	1.2905	↑
55	Pos	tert-butyl(2S)-2-[4-[[(2S)-1-[(4-methoxyphenyl) methylamino]-3-methyl-1-oxobutan-2 yl] carbamoyl] piperidine-1-carbonyl] pyrrolidine-1-carboxylate	C_29_H_44_N_4_O_6_	10.212	545.3418	M + H	1.2298	↑
56	Pos	PE (21:4)	C_26_H_44_NO_8_P	4.722	552.2757	M + Na	1.8464	↓
57	Pos	LysoPC (19:0)	C_27_H_56_NO_7_P	11.590	560.3672	M + Na	1.1769	↑
58	Pos	LysoPC (22:0)	C_30_H_62_NO_7_P	12.543	580.4318	M + H	2.2871	↑
59	Pos	Ouabain	C_29_H_44_O_12_	4.949	585.2894	M + H	1.5374	↓
60	Pos	PI (19:1(9Z)/0:0)	C_28_H_53_O_12_P	5.048	613.3411	M + H	1.3098	↓
61	Pos	SM (d33:1)	C_38_H_77_N_2_O_6_P	13.386	689.5570	M + H	1.7413	↑
62	Pos	PC (36:2)	C_44_H_84_NO_8_P	12.566	786.5980	M + H	1.4118	↑
63	Pos	PC (38:3)	C_46_H_86_NO_8_P	11.852	812.6132	M + H	1.382	↑
64	Pos	PC (40:4)	C_48_H_88_NO_8_P	11.823	838.6292	M + H	1.4059	↑
65	Pos	Tunicamycin B	C_39_H_64_N_4_O_16_	5.433	845.4148	M + H	1.2985	↓
66	Pos	TAG (64:19)	C_67_H_92_O_6_	9.977	1015.6696	M + Na	1.3547	↑

**Table 2 ijms-23-07978-t002:** Metabolic pathways of radiation-injured mice on potential biomarkers.

Pathnames	PathIds	Total	Hits	Raw *p*	−log (*p*)	Holm Adjust	FDR	Impact
Glycerophospholipid metabolism	mmu00564	36	3	0.02475	3.6989	1.0	1.0	0.25169
Nicotinate and nicotinamide metabolism	mmu00760	15	2	0.028158	3.5699	1.0	1.0	0.1943
Sphingolipid metabolism	mmu00600	21	2	0.052726	2.9426	1.0	1.0	0.15822
Linoleic acid metabolism	mmu00591	5	1	0.086595	2.4465	1.0	1.0	0
Phosphatidylinositol signaling system	mmu04070	28	2	0.087932	2.4312	1.0	1.0	0.03888
Ascorbate and aldarate metabolism	mmu00053	10	1	0.16595	1.7961	1.0	1.0	0
alpha-Linolenic acid metabolism	mmu00592	13	1	0.21032	1.5591	1.0	1.0	0
Glycerolipid metabolism	mmu00561	16	1	0.25242	1.3766	1.0	1.0	0.01246
Galactose metabolism	mmu00052	27	1	0.38906	0.94401	1.0	1.0	0
Inositol phosphate metabolism	mmu00562	30	1	0.42194	0.8629	1.0	1.0	0.12939
Glycine, serine and threonine metabolism	mmu00260	34	1	0.46313	0.76975	1.0	1.0	0.02408
Arachidonic acid metabolism	mmu00590	36	1	0.48265	0.72847	1.0	1.0	0
Biosynthesis of unsaturated fatty acids	mmu01040	36	1	0.48265	0.72847	1.0	1.0	0
Arginine and proline metabolism	mmu00330	38	1	0.50149	0.69018	1.0	1.0	0.02346
Tryptophan metabolism	mmu00380	41	1	0.52851	0.6377	1.0	1.0	0.14305
Aminoacyl-tRNA biosynthesis	mmu00970	48	1	0.58618	0.53412	1.0	1.0	0

## Data Availability

The raw data are available on request from the corresponding author.

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
