# Peer review of "UHPLC/MS-Based Serum Metabolomics Reveals the Mechanism of Radiation-Induced Thrombocytopenia in Mice"

_ijms, 2022, doi:10.3390/ijms23147978_

Round 1

Reviewer 1 Report

Dear Authors,

Please add more conclusion to your research

Kind regards

Author Response

Thanks a lot for the excellent comments and suggestions. As you suggested, we have added more conclusions to make the content more solid and credible (page 17, line 490-493 and line 502-506). And the manuscript has undergone English language editing.

Reviewer 2 Report

The authors made some relevant effort to address the recovery phase of the question, although not optimal, it is acceptable.

Author Response

Thanks a lot for the excellent comments. We added some relevant references in the introduction to enrich the background of the article (page 2, line 50-55). At the same time, we deleted references that were not very relevant to the article. And the manuscript has undergone English language editing.

This manuscript is a resubmission of an earlier submission. The following is a list of the peer review reports and author responses from that submission.

Round 1

Reviewer 1 Report

This manuscript, by Xiong, L et.al., et al., tried to understand the mechanisms of radiation-induced thrombocytopenia in mice by using UHPLC/MS-Based serum metabolomics.  The authors also applied the pathway analysis method to couple with the metabolomics to reveal the dysregulated glycerol-phospholipid metabolism shown by increased or decreased accumulation of some metabolites of these pathways and concluded that these metabolites directly or indirectly lead to inhibition of megakaryocyte differentiation which results to thrombocytopenia.

This reviewer thinks the topic of the manuscript is important, the understanding of the mechanisms of radiation-induced thrombocytopenia could lead to better clinical target and outcome for thrombocytopenia management.  However, there are some major issues this manuscript needs to fix: 

  • Experimental design: it is known that in mouse, day 11 post irradiation is the lowest point for platelet count and by day 30, the platelet count is recovered to normal (Tkaczynski, E et. Al., PMID: 29599195). The authors compared non-radiated to day 11 radiated mice to get the data represented here, which is good.  However, the other half of the story is not answered: at day 30, when the mice platelet number recovers, are those metabolites also returned to normal?  Or only some of those returned to normal?  Based on this, which metabolites would be more important in regulating megakaryocyte differentiation?
  • Radiation alone can cause cell damage. Data presented in this manuscript reveals the association of changes of metabolites and cell malfunction.  The question “does cell damage release abnormal metabolites or the abnormal metabolites cause cell damage” is a chicken-and-egg question, the set up testing system in this manuscript is not sufficient to answer it, thus the conclusion of the manuscript is not solid
  • Authors mentioned in methods section that they used both male and female mice in the study which is good. Please present the data separately with n number for each group and treatment
  • Why using K562 – a human cell line to test the metabolites’ function, not a murine megakaryocyte stem cell/line?

Some minor critics are as following:

  • P2, line 46, “organisms” should be “organs”
  • P2, “Results” section, please detail sex, n number, treatment here in addition to the methods section
  • P3, line 115, “remarkably” should be “markedly”
  • P6, line 160, “up-graded and down-graded” should be “up-regulated and down-regulated”
  • P13, line 249, “66” should be spelled out as “Sixty-six”. Don’t start a sentence with numerical numbers; same with p14, line 333, number “32” should be spelled out
  • P15, line 341, “random” should be “randomly”

Reviewer 2 Report

Dear Authors,

In my opinion, it is very relevant and interesting, well written, novelty, and easy to read. I think the authors must add more conclusions. 

Please add more conclusions to your research

Please add more references

Kind regards

Round 2

Reviewer 1 Report

The authors answered all of my questions on paper, but some needed experiments are not done.  I don't think this revision improved very much aside by adding some discussions.

Round 3

Reviewer 1 Report

The newly proposed experiment (point 1) by the authors will satisfy the main concerns of this reviewer.